# In Silico Screening of Novel TMPRSS2 Inhibitors for Treatment of COVID-19

**DOI:** 10.3390/molecules27134210

**Published:** 2022-06-30

**Authors:** Shuo Wang, Xuexun Fang, Ye Wang

**Affiliations:** Key Laboratory for Molecular Enzymology and Engineering of Ministry of Education, School of Life Sciences, Jilin University, 2699 Qianjin Street, Changchun 130012, China; ws20@mails.jlu.edu.cn (S.W.); fangxx@jlu.edu.cn (X.F.)

**Keywords:** SARS-CoV-2, COVID-19 drugs, virtual screening, transmembrane protease serine 2 (TMPRSS2), molecular dynamics simulation

## Abstract

COVID-19, a pandemic caused by the virus SARS-CoV-2, has spread globally, necessitating the search for antiviral compounds. Transmembrane protease serine 2 (TMPRSS2) is a cell surface protease that plays an essential role in SARS-CoV-2 infection. Therefore, researchers are searching for TMPRSS2 inhibitors that can be used for the treatment of COVID-19. As such, in this study, based on the crystal structure, we targeted the active site of TMPRSS2 for virtual screening of compounds in the FDA database. Then, we screened lumacaftor and ergotamine, which showed strong binding ability, using 100 ns molecular dynamics simulations to study the stability of the protein–ligand binding process, the flexibility of amino acid residues, and the formation of hydrogen bonds. Subsequently, we calculated the binding free energy of the protein–ligand complex by the MM-PBSA method. The results show that lumacaftor and ergotamine interact with residues around the TMPRSS2 active site, and reached equilibrium in the 100 ns molecular dynamics simulations. We think that lumacaftor and ergotamine, which we screened through in silico studies, can effectively inhibit the activity of TMPRSS2. Our findings provide a basis for subsequent in vitro experiments, having important implications for the development of effective anti-COVID-19 drugs.

## 1. Introduction

The COVID-19 pandemic caused by the RNA virus SARS-CoV-2 has severely impacted public health and the economy worldwide [1]. Its transmission is relatively strong, and the Omicron variant is still prevalent in China and around the world. SARS-CoV-2 belongs to the β genus of coronaviruses and has an envelope, round or oval particles, and a genome size of 29.9 KB [2,3]. It has five essential genes encoding a variety of structural and nonstructural proteins. The main structural proteins of the virus particles include the spike (S), envelope (E), matrix (M), and nuclear (N) proteins [4]. The nucleoprotein (N) wraps the RNA genome to form the nucleocapsid, which is surrounded by the viral envelope (E). The virus envelope contains the matrix (M) and spike (S) proteins. The spike protein (S) of SARS-CoV-2 is a critical protein enabling the viral infection of host cells [5].

The two most commonly used approaches to fighting COVID-19 infection are vaccine prophylaxis and drug therapy. Many types of vaccines are available, including inactivated virus, protein, adenovirus vector, and mRNA vaccines. These vaccines have been approved by the U.S. Food and Drug Administration (FDA) under emergency authorization and are helpful in reducing COVID-19 mortality and morbidity. However, some outstanding issues remain, such as the effectiveness of the defense against the virus provided by vaccines, the duration of protection, and the continued mutation of the virus [6], which may require the continued preparation of new vaccines or vaccine boosters to combat emerging variants [7]. These new vaccines have substantially reduced COVID-19 infection and mortality, but using vaccines alone is not a viable strategy to overcome this pandemic [8,9]. Therefore, drug treatment is a practical solution. Some important drug targets have been found: viral proteins and some host proteins, such as the viral spike protein, host cell ACE2 receptor, and TMPRSS2 active site [6].

When SARS-CoV-2 invades a host cell, TMPRSS2, which is located on the cell membrane, cleaves the S2′ site in the S2 spike protein, exposing the fusion peptide (FP) and allowing the virus to enter the host cell [10,11]. In animal models of SARS and MERS coronavirus infections, the knockout of TMPRSS2 considerably reduces respiratory transmission [12]. The TMPRSS2 inhibitor nafamostat mesylate substantially blocks SARS-CoV-2 infection of lung cells [13]. Therefore, TMPRSS2 is crucial and may be used as a target for drug screening [14].

TMPRSS2 is a transmembrane serine protease widely expressed in the epithelial cells of the respiratory, gastrointestinal, and urogenital tracts [15], and is highly expressed in the prostate and colon [16]. The TMPRSS2 gene encodes 492 amino acids that comprise the self-membrane protein [17]. Mutating Ser441 causes TMPRSS2 to become an inactive mutant, so this residue is necessary [18].

Computer-aided drug design (CADD) has been increasingly used in novel drug development, to help speed up the design of and reduce the funds required for novel drug development. Given the high failure rate of clinical studies of new drugs, finding new indications for existing drugs has become an attractive strategy to minimize drug development costs [19,20,21,22,23]. As such, in this study, we screened TMPRSS2 active site small-molecule inhibitors from the FDA that may inhibit TMPRSS2 activity, thereby blocking SARS-CoV-2 entry into cells and providing a possible therapeutic approach for COVID-19. From the screening results, we selected lumacaftor and ergotamine, which showed strong binding capacity, for 100 ns molecular dynamics simulations to study the stability of the protein–ligand binding process, flexibility of amino acid residues, and formation of hydrogen bonds. Then, we calculated the free energy of the small molecule–protein complex by the MM-PBSA method. The results show that lumacaftor and ergotamine interact with residues around the TMPRSS2 active site, and reached equilibrium in the 100 ns molecular dynamics simulations. Because of these properties, we think that lumacaftor and ergotamine, which we screened in our in silico studies, can effectively inhibit the activity of TMPRSS2. Our findings provide a basis for subsequent in vitro experiments and have important implications for the development of effective anti-COVID-19 drugs.

## 2. Results

### 2.1. TMPRSS2 Homology Modeling and Virtual Screening

Because some amino acids of the crystal structure of TMPRSS2 were missing, we reconstructed its three-dimensional structure using the homology model method. The reconstructed structure and active center are shown in Figure 1. Subsequently, we compared the structure of the reconstructed TMPRSS2 with that of 7MEQ in the RCSB Protein Data Bank Database (PDB) and found that the structure of the reconstructed TMPRSS2 was similar to that in the PDB, with an RMSD of only 0.049 Å (Figure 1B).

After virtual screening of the FDA database based on the structure of TMPRSS2, we ranked the 1410 compounds that interact with TMPRSS2 by affinity score. Appendix A shows the details of the top 10 compounds, as well as the known TMPRSS2 inhibitors nafamostat and camostat. All top 10 compounds (<−8.6 kcal/mol) had a binding energy to TMPRSS2 lower than that of nafamostat (−6.1 kcal/mol). The superposition of nafamostat between the lowest bound free-energy conformation and the crystal structure is shown in Appendix A, and the RMSD between them is 0.508 Å (Appendix A).

### 2.2. Interaction Residues of TMPRSS2 with Potential Inhibitors

By analyzing the hydrogen bond interaction between TMPRSS2 and the top 10 compounds, we found that these compounds all bind to the residues around the active center of TMPRSS2, such as His296, Lys342, Cys390, Ser436, Cys437, Gly439, Ser441, Trp461, Gly462, Ser463, and Gly464 (Figure 2). Ser441, an important residue at the active site, can hydrogen bond to all eight compounds except dihydroergotamine and glecaprevir. Appendix A provide detailed information such as the residues of TMPRSS2 that interact via hydrogen bonds with potential inhibitors. Ansamycin is a class of antibiotics, and this particular molecule is not currently used in the pharmaceutical industry. Therefore, we selected the drugs ergotamine and lumacaftor, which have equal secondary binding energy, for further molecular dynamics analyses.

### 2.3. TMPRSS2 and Ergotamine or Lumacaftor Conformations Retain Stability during MD Simulations

We performed molecular dynamics simulations of the TMPRSS2–ergotamine system and the TMPRSS2–lumacaftor system for 100 ns to understand their binding modes. We repeated the simulations three times. The data are shown in Appendix A. We acquired the initial conformation of the inhibitor from the optimal pose of the virtual screening. Because the RMSD value is an essential parameter used for assessing the convergence equilibrium of a system, we separately calculated the RMSD values for TMPRSS2 (in systems bound to ergotamine and lumacaftor) and the ligands (ergotamine and lumacaftor). Figure 3A shows that a stable equilibrium state was reached for both protein and ligand conformations during the 100.0 ns simulations. We also obtained an identical conclusion from the analysis of the radius of gyration (Rg) of the protein (Figure 3B). We evaluated the flexibility of the protein conformation by calculating the RMSF value, and we concluded that the whole protein conformation was stable (Figure 3C). The amino acid residues that were more flexible were far from the active pocket (Figure 3D). During the simulation, the binding regions of ergotamine and lumacaftor to the protein were identified, as shown in Figure 3E,F.

Subsequently, we analyzed the number of hydrogen bonds between TMPRSS2 and each ligand (ergotamine and lumacaftor) as well as their center-of-mass distance in the simulation. As shown in Figure 4A,B, during the 100 ns simulation, TMPRSS2 formed four and three hydrogen bonds with lumacaftor and ergotamine, respectively. As shown in Figure 4C, the center-of-mass distance between the Ser441 residue of TMPRSS2 was 1.0 ± 0.1 nm with ergotamine and 1.4 ± 0.1 nm with lumacaftor during the simulation. This indicates that the center-of-mass distance between the Ser441 residue of TMPRSS2 and ergotamine or lumacaftor essentially remained stable.

### 2.4. Interaction Mechanism between TMPRSS2 and Ergotamine or Lumacaftor

Subsequently, we categorized the structures of the protein–ligand complexes with similar conformations into identical clusters during the 100 ns molecular simulation analysis. We found eight clusters in both protein–ligand complexes. Then, we observed the ligand–protein binding of the representative structure of each cluster, with the results shown in Appendix A. Then, we extracted the representative frames from the largest clusters for analysis. Figure 5A,B show that the ligands ergotamine and lumacaftor can each bind to TMPRSS2, forming hydrogen bond and hydrophobic interactions, respectively. Figure 5A,B show that the Lys342, Glu389, Trp416, Leu419, Ile420, Gln438, Trp461, Gly462, Gly464, Ser436, Ser463, Cys465, and Arg470 residues are involved in the binding.

### 2.5. Calculation of MM-PBSA Binding Free Energy between TMPRSS2 and Ergotamine or Lumacaftor

We used the MM-PBSA method to accurately calculate the binding free energy of the protein–ligand complexes [24]. The final binding free energy is the sum of the contributions of the van der Waals force, electrostatic force, polar solvation, and solvent accessible surface area (SASA) energy components. The contribution of each of the above components to the binding energy is shown in Table 1. Due to the stability of the system, we chose the first 30 ns for the MM-PBSA calculation and found in the simulation process that lumacaftor and ergotamine tightly bind to TMPRSS2.

## 3. Discussion

COVID-19, which is caused by SARS-CoV-2, has spread worldwide, so workable solutions in addition to vaccination are required. Although several types of vaccines have been developed, the constant mutation of the RNA virus has led to vaccine inefficiency. Identifying potent drugs based on specific targets is a more reliable and effective solution for treatment of COVID-19 infection. In December 2021, the FDA authorized the first oral antiviral drug, Paxlovid (copackaged nimatoprevir and ritonavir), for the treatment of COVID-19. In particular, nirmatrelvir (PF-07321332) is a potent and selective inhibitor of the main protease (M^PRO^) of SARS-CoV-2, which plays a critical role in viral replication [25]. TMPRSS2 is a transmembrane serine protease involved in the fusion process of ACE2 and SARS-CoV-2 membranes and is necessary for virus invasion of the host, so it is a target for drug screening.

In this study, we used the crystal structure of TMPRSS2 as a virtual screening target to identify potentially effective drugs for COVID-19 treatment from the FDA database. Subsequently, we identified two potent inhibitors, namely, lumacaftor and ergotamine. We indirectly evidenced the inhibition of TMPRSS2 by lumacaftor or ergotamine through a series of molecular dynamics analyses, including protein–ligand binding stability, binding free energy, hydrogen bond number, and center-of-mass distance. It has long been proven that ergotamine has the ability to bind with the SARS-CoV-2 M^PRO^, PL^PRO^, S protein, RdRp protein, and 2′-O-MTase, and human NRP1 [26,27,28,29,30,31,32]. Lumacaftor also has the ability to bind with the S protein [33,34].

Ergotamine has some side effects, as does lumacaftor. Common adverse side effects of ergotamine include irritation, nausea, vomiting, headache, diarrhea, tingling in the limbs, and confusion. The main side effect of ergotamine (alpha-1 adrenergic agonist) is vasoconstriction, which can be dangerous in severe cases of COVID-19. Lumacaftor also has some adverse side effects: dyspnea, abnormal breathing, and upper respiratory infections are common. These two compounds may be promising for COVID-19 but have serious side effects. Therefore, we propose them as lead compounds to improve both their selectivity and pharmacokinetic properties.

In this study, we screened two potent TMPRSS2 inhibitors, lumacaftor and ergotamine, from the FDA database by a computer-aided drug design approach. The findings suggest that lumacaftor and ergotamine can be developed as COVID-19 therapeutic agents. The drug repurposing of lumacaftor and ergotamine can improve the success rate of novel drug development. Finally, the approach we applied in this study contributes to the fight against this pandemic by providing a rapid and feasible strategy to develop an effective drug for COVID-19 infection treatment.

## 4. Materials and Methods

### 4.1. Protein TMPRSS2 Structure Preparation

We obtained the crystal structure of the protein TMPRSS2 (PDB ID 7MEQ) [35] from the RCSB Protein Data Bank database (https://www.rcsb.org/, accessed on 1 April 2022) [36]. Because some amino acid residues of TMPRSS2 had not been located, we performed automatic homology modeling to reconstruct the structure through the online server Swiss-Model (https://swissmodel.expasy.org/, accessed on 1 April 2022) [37]. We displayed and analyzed the protein structure using the Pymol 2.5 visualization program [38,39,40].

### 4.2. Virtual Screening

We downloaded 1410 compounds in mol2 format from the FDA-approved drug database (https://zinc15.docking.org/catalogs/subsets/fda/, accessed on 1 April 2022) for virtual screening [41]. We used Raccoon 1.0 to convert the mol2 files of compounds into PDBQT files that could be identified by virtual screening software [42]. We performed virtual screening using the AutoDock Vina v.1.2.0 program, which provides rapid one-step docking analysis [43]. We set the parameter values of the x, y, and z centers to −9.212, −6.199, and 19.840, and set the parameter values of the grid map in the x, y, and z dimensions to 26 × 26 × 26 Å.

### 4.3. Molecular Dynamics (MD) Simulation

We used MD to simulate the process through which the protein complexes with the ligand using Gromacs version 2022 with the CHARMM36 Force Field [44]. We obtained the initial conformation of the ligand from the optimal pose of the virtual screening. We generated the force field of the ligand through the CGenFF server, which is the official CHARMM General Force Field server [45,46].

The protein–ligand complexes were located in the center of the cubic cell and were solubilized using the TIP3P water model. Because the total charge of the complex with the ligand ergotamine was 3.000 e, we added three additional CL ions to the system to automatically achieve electro-neutrality. However, we added no extra ions to the system because the complex with the ligand lumacaftor was already electrically neutral. The conformation was relaxed by energy minimization to ensure a reasonable starting structure in both geometry and solvent orientation. Convergence was achieved at a maximum force of less than 1000 kJ/mol·nm in any atom. We considered two equilibrium phases, the NVT ensemble (constant number of particles, volume, and temperature) and NPT ensemble (constant number of particles, pressure, and temperature), each for 100 ps, until the system reached equilibrium. We performed dynamics simulations for 100 ns after equilibration. We calculated the root mean square deviation (RMSD), root mean square fluctuation (RMSF), and radius of gyration (Rg) using the built-in utilities in the Gromacs package. The number of hydrogen bonds between a protein and ligand and their distances can also be calculated using the Gromacs internal module. Gromacs includes a cluster analysis tool, which can be used to classify the conformation based on the RMSD, which we set to a cutoff value of 2.5 Å. We displayed, animated, and analyzed protein trajectories using the Pymol visualization program [38,39,40]. We used Ligplot to analyze the interaction between the protein and ligands [47].

### 4.4. MM-PBSA Calculation

We calculated the binding free energy of TMPRSS2 with lumacaftor and ergotamine using the molecular mechanics Poisson–Boltzmann surface area (MM-PBSA) method, which considers the solvent as a homogeneous continuous medium and averages multiple frames in the equilibrium trajectory [48]. The binding free energy is defined as [49]
ΔG_bind,aq_ = ΔH − TΔS ≈ ΔE_MM_ + ΔG_bind,solv_ − TΔS,(1)
ΔE_MM_ = ΔE_covalent_ + ΔE_electrostatic_ + ΔE_vdW_,(2)
ΔE_covalent_ = ΔE_bond_ + ΔE_angle_ + ΔE_torsion_,(3)
ΔG_bind,solv_ = ΔG_polar_ +ΔG_non__-polar_,(4)

GROMACS also includes an internal tool, g_mmpbsa, to calculate the binding free energy directly from the trajectory. We used g_mmpbsa to calculate the 100 ns trajectory obtained by MD simulation, and we selected the 0–30 ns trajectory after the system was stabilized to calculate the binding energy of the ligand and protein.

## Figures and Tables

**Figure 1 molecules-27-04210-f001:**
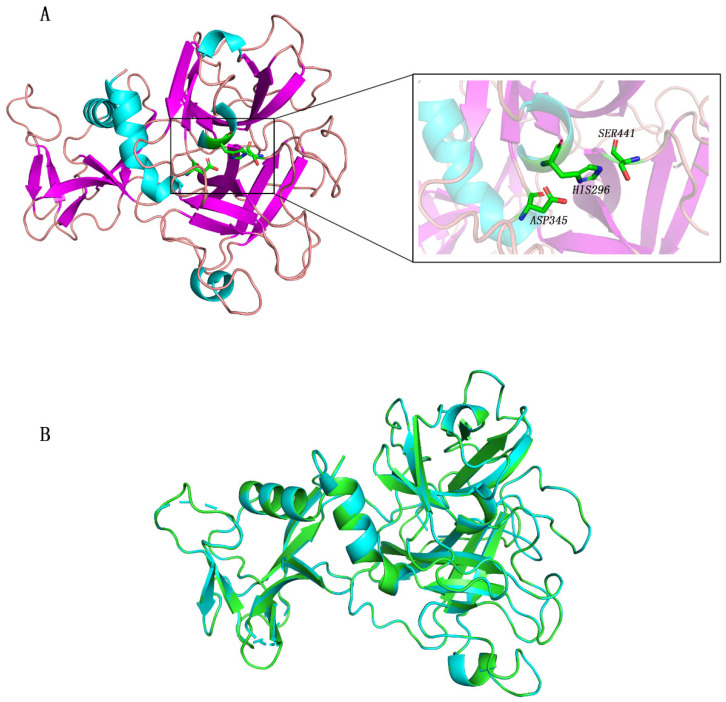
Structure of TMPRSS2 (PDB ID 7MEQ). Active site residues His296, Asp345, and Ser441 are presented as green sticks (**A**). Alignment diagram between 7MEQ and reconstructed TMPRSS2; RMSD is 0.049 Å (**B**). TMPRSS2 is presented in green; 7MEQ is presented in blue.

**Figure 2 molecules-27-04210-f002:**
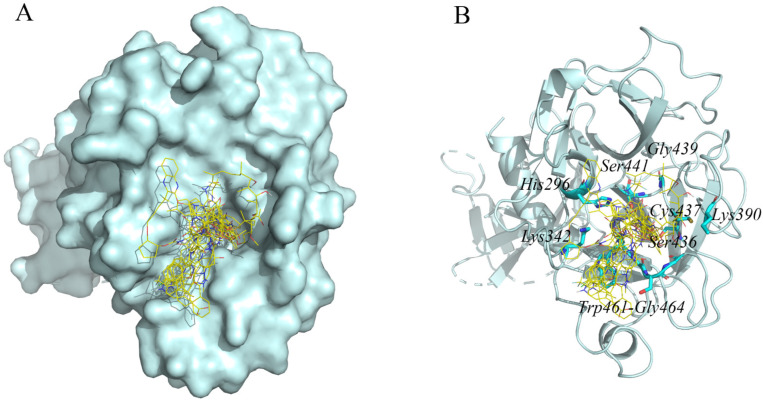
Diagram of the top 10 compounds binding to the protein TMPRSS2. (**A**) The top 10 compounds are presented as yellow lines; TMPRSS2 is presented in cyan. (**B**) The top 10 compounds are presented as yellow lines; TMPRSS2 is presented as a cyan cartoon; and amino acid residues of TMPRSS2 bound to the inhibitor are presented as cyan sticks.

**Figure 3 molecules-27-04210-f003:**
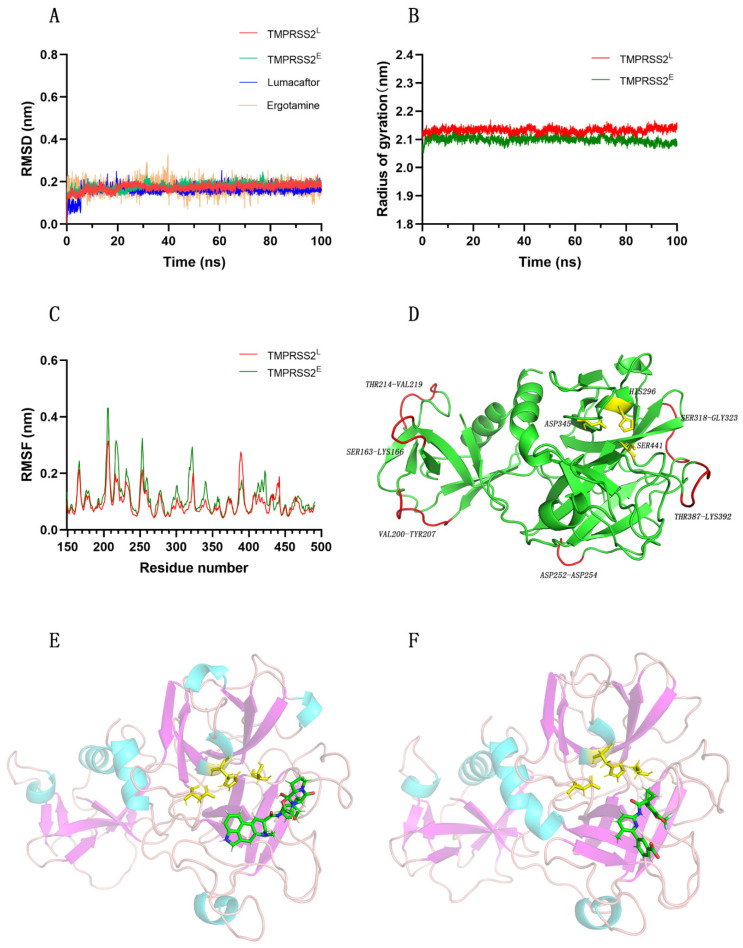
Schematic representation of RMSD (**A**), radius of gyration (**B**), and RMSF (**C**) of the systems during 100 ns MD simulations. (**D**) Flexible amino acid residues and active sites during the 100 ns MD simulation. TMPRSS2E/L represents the protein structure of the complex of TMPRSS2 with ergotamine or lumacaftor. Structures in red are amino acid residues with increased freedom, while bars in yellow represent active sites of TMPRSS2. Binding region of ergotamine (**E**) or lumacaftor (**F**) with TMPRSS2.

**Figure 4 molecules-27-04210-f004:**
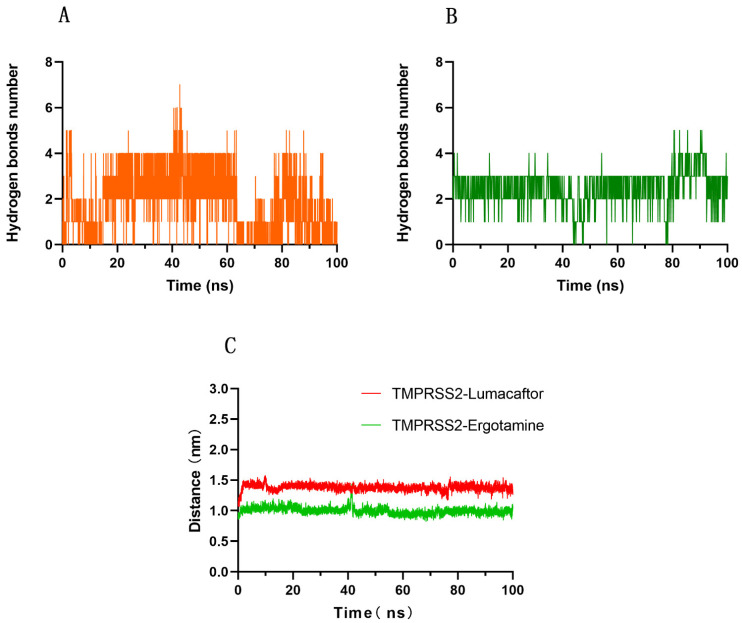
Number of hydrogen bonds between (**A**) lumacaftor or (**B**) ergotamine and TMPRSS2. Center-of-mass distance (**C**) between Ser441 residue of TMPRSS2 and ergotamine or lumacaftor.

**Figure 5 molecules-27-04210-f005:**
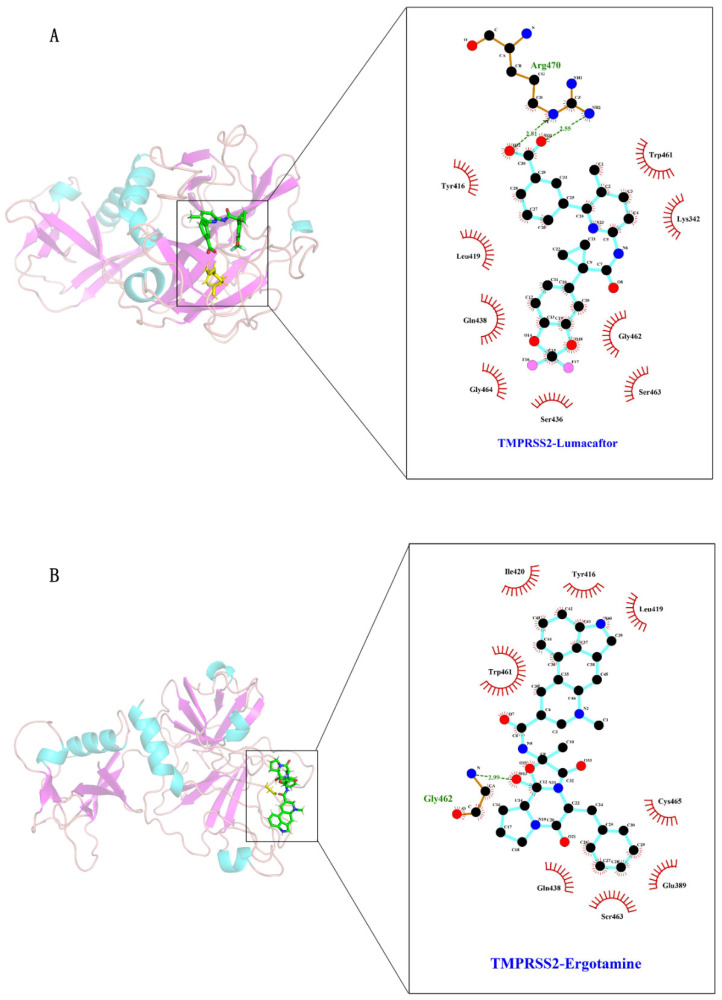
Interaction of amino acid residues around the active site of TMPRSS2 with inhibitors. Inhibitors lumacaftor (**A**) and ergotamine (**B**) are represented as green sticks. TMPRSS2 is represented by a cartoon structure. Amino acid residues hydrogen-bonded to inhibitors are shown as yellow sticks. Dotted green lines represent hydrogen bonds, and numbers indicate the distance between donor and acceptor atoms. Red circles represent hydrophobic interactions.

**Table 1 molecules-27-04210-t001:** Binding free energy of TMPRSS2 with lumacaftor and ergotamine based on calculations using the MM-PBSA method.

Protein–Ligand Complex	Binding Energy (kcal/mol)	Van Der Waals Energy (kcal/mol)	Electrostatic Energy (kcal/mol)	Polar Solvation Energy (kcal/mol)	SASA Energy (kcal/mol)
TMPRSS2–Lumacaftor	−16.764 ± 3.553	−31.828 ± 2.904	−16.391 ± 2.739	35.249 ± 6.915	−3.775 ± 0.314
TMPRSS2–Ergotamine	−17.678 ± 3.718	−42.002 ± 3.117	−10.380 ± 2.166	39.232 ± 3.572	−4.528 ± 0.257

## Data Availability

Not applicable.

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
