# Peer review of "In Silico Screening of Novel TMPRSS2 Inhibitors for Treatment of COVID-19"

_molecules, 2022, doi:10.3390/molecules27134210_

Round 1

Reviewer 1 Report

Dear authors,

Generally a very well organized and written paper with just few unclear parts:

1) It is not clear to the reader how you ended up with Lumacaftor & Ergotamine. Is it from your screening or from literature data? Please elaborate a bit more here to support their TMPRSS2 targeting, i.e. reference?

2) In the database you used did you apply any further filtering or you screened all 1200+ drugs?

3) Please provide an RMSD value value between the reconstructed TMPRSS2 structure and 7MEQ

Best regards

Author Response

Point 1: It is not clear to the reader how you ended up with Lumacaftor & Ergotamine. Is it from your screening or from literature data? Please elaborate a bit more here to support their TMPRSS2 targeting, i.e. reference?

Response 1: We thank the reviewer for this critical question. Lumacaftor & Ergotamine was obtained by screening 1,410 small molecule compounds. It is mentioned in this article in lines 91-95 and 109-112.

Lines 91-95 read as follows:

“After virtual screening of the FDA database based on the structure of TMPRSS2, the 1410 compounds that interacted with TMPRSS2 were ranked by affinity score. Table S1 shows the details of the top 10 compounds as well as the known TMPRSS2 inhibitors Nafamostat and Camostat. All top 10 compounds (<-8.6kcal/mol) were lower than Nafa-mostat (-6.1kcal/mol) in terms of binding energy to TMPRSS2.”

Lines 109-112 read as follows:

“Ansamycin is a class of antibiotics, and this particular molecule is not currently used in the pharmaceutical industry. Therefore, the drugs Ergotamine and Lumacaftor, which had the second binding energy equally, were selected for the further molecular dynamics analysis.”

Point 2: In the database you used did you apply any further filtering or you screened all 1200+ drugs?

Response 2: We are grateful to the reviewer for pointing out this issue. In the database I used, all 1,410 drugs were screened. It is mentioned in lines 91-95 of this article.

 “After virtual screening of the FDA database based on the structure of TMPRSS2, the 1410 compounds that interacted with TMPRSS2 were ranked by affinity score. Table S1 shows the details of the top 10 compounds as well as the known TMPRSS2 inhibitors Nafamostat and Camostat. All top 10 compounds (<-8.6kcal/mol) were lower than Nafa-mostat (-6.1kcal/mol) in terms of binding energy to TMPRSS2.”

Point 3: Please provide an RMSD value value between the reconstructed TMPRSS2 structure and 7MEQ.

Response 3: We deeply appreciate the reviewer’s suggestion. We have added align images of reconstructed TMPRSS2 and 7MEQ in Figure 1B, and marked RMSD between them as 0.049 in the annotations in lines 99-100.

“Align diagram between 7MEQ and reconstructed TMPRSS2, RMSD is 0.049 Å (B). The TMPRSS2 is represented as green cartoon structure and the 7MEQ is represented as blue cartoon structure.”

Reviewer 2 Report

The authors present the search for novel inhibitors for transmembrane protease serine 2 (TMPRSS2) by repurposing FDA approved small molecules. This is an interesting target and a timely contribution, as TMPRSS2 processes Spike rendering it competent to promote membrane fusion in SARS-CoV2 infection. The modeling approach is straightforward and follows standard procedures, yet there are important details that require attention before this is ready to be published:

It may be a style issue, but the authors claim to have identified the active site of the enzyme. That information is available in the PDB entry they used for their work, and also in the UNIPROT entry for the protein. Therefore, the authors did not identify the active site residues. They are already known. This also makes lines 64 and 89 redundant. Also regarding the active site, the authors cite a study where Ser441 was mutated, leading to loss of "some function". The immediate sentence states that therefore this residue is essential; that means that all function is lost. Either it is essential or some function is lost. 

Regarding docking, in lines 101-110, as binding was restricted to the active site, the ligands had no other place to go. The issue there is how the ligands chose to bind. Also, it is common practice to validate the docking protocol by performing a redocking of a known ligand (in this case, the one found in the crystal structure). The data for the binding energy is reported in the supplementary table, yet, a superposition of the predicted and experimental poses should be shown (maybe as another supplementary figure).

Figures 3A and B show that the complexes are stable, but the ligands are not: there are at least two states for lumacaftor and three for ergotamine. This could explain why on figure 3C the RMSFs are larger for the ergotamine complex. The authors seem to be calculating RMSFs using only one structure as reference for the whole simulation, instead of using a particular reference for each of the states for the ligand. Panel 3D is not very informative; I would use it to show the zones in the protein that respond differently to each ligand.

Figure 4A shows that indeed the COM distance is very stable, yet Figure 3A shows that there are different conformations for the ligands, and figure 4B shows loss of interactions. I conclude that COM is not a very informative property.

Regarding figure 5, the authors should include in their analyses other interactions, not just H bonds. I suggest using LigPlot over the same representative structures. That would highlight other relevant interactions that would then help explain the differences found in the MM-PBSA calculations. About the clustering (line 152), how many clusters were found? Are there any relevant differences, regarding ligand binding, amongst these structures?

In Table 2, the rather large standard deviations suggest  heterogeneity in the structures used for the calculation, in agreement with Figure 3A. That is not a sound protocol for MM-PBSA calculations.

In the methods section, regarding the preparation of the structure, indeed, there are large gaps missing in the crystal structure. Reconstruction with SwissModel is a very good idea. On the other hand, there is a functional cleavage at residues 255 and 256, as stated in the UNIPROT entry for this protein, and I wonder if the authors took this into account. This processing can be important for the local electrostatics, as in generates extra N- and C-termini in the protein that could lie close to the active site, influencing ligand binding indirectly.

Regarding the MD simulations, a single 100 ns MD run to explore stability is not enough. See as a reference the work of Wonpil Im, where at least three runs are needed. Also, I would like to know why the SPC216 water model was used with CHARMM36, instead of the recommended TIP3P. Simulations with no added salt may lead to artificial increases in electrostatic interactions; as this is an extracellular protein, I suggest adding 0.15M NaCl.

Finally, regarding MM-PBSA, I would like the authors to explain how the TS contribution is calculated, and also to state which section of the 100 ns run was used for the calculation. As I stated above, using the whole run is not adequate, due to the data for the ligands shown in figure 3A.

Author Response

Point 1: It may be a style issue, but the authors claim to have identified the active site of the enzyme. That information is available in the PDB entry they used for their work, and also in the UNIPROT entry for the protein. Therefore, the authors did not identify the active site residues. They are already known. This also makes lines 64 and 89 redundant. Also regarding the active site, the authors cite a study where Ser441 was mutated, leading to loss of "some function". The immediate sentence states that therefore this residue is essential; that means that all function is lost. Either it is essential or some function is lost.

Response 1: We are grateful for the suggestion. As suggested by the reviewer, we have deleted line 64 and modified lines 85-87 and lines 63-64.

Lines 85-87 reads as follows:

“By analyzing the structural interaction between TMPRSS2 and Nafamostat, we determined the active site of TMPRSS2 and presented it in Figure 1A.”

Lines 63-64 reads as follows:

“It has previously been shown that mutating Ser441 causes TMPRSS2 to become an inactive mutant, so this residue is necessary.”

Point 2: Regarding docking, in lines 101-110, as binding was restricted to the active site, the ligands had no other place to go. The issue there is how the ligands chose to bind. Also, it is common practice to validate the docking protocol by performing a redocking of a known ligand (in this case, the one found in the crystal structure). The data for the binding energy is reported in the supplementary table, yet, a superposition of the predicted and experimental poses should be shown (maybe as another supplementary figure).

Response 2: We are very grateful to the reviewers for their questions. As suggested by the reviewer, we have added the docking diagram and crystal structure comparison of TMPRSS2 and Nafamostat into the supplementary data. It is described in lines 95-96 of the article.

“The contrast between Nafamostat and TMPRSS2 docking picture and crystal structure is shown in Figure S1.”

Point 3: Figures 3A and B show that the complexes are stable, but the ligands are not: there are at least two states for lumacaftor and three for ergotamine. This could explain why on figure 3C the RMSFs are larger for the ergotamine complex. The authors seem to be calculating RMSFs using only one structure as reference for the whole simulation, instead of using a particular reference for each of the states for the ligand. Panel 3D is not very informative; I would use it to show the zones in the protein that respond differently to each ligand.

Response 3: We are very grateful to the reviewers for their questions. We are very sorry that we did not indicate clearly in Figure 3A, which caused your misunderstanding. In Figure 3A, RMSD of single ligand and single protein is not RMSD of ligand and protein complex. We have re-marked the legend in the Figure. According to your suggestion, we conducted another two MD simulations, and the result was that there was only one conformation of the ligand, which we have shown in Figure 3A. Therefore, the difference in RMSF may be influenced by other factors. Then we showed the ergotamine /Lumacaftor and protein reaction regions in Figure 3E and Figure 3F.

Point 4: Figure 4A shows that indeed the COM distance is very stable, yet Figure 3A shows that there are different conformations for the ligands, and figure 4B shows loss of interactions. I conclude that COM is not a very informative property.

Response 4: Since the previous result was accidental, we repeated the experiment according to your suggestion, and found that RMSD of the protein and ligand were stable.  We recalculated the center-of-mass distance of the inhibitor and Ser441 residues of the protein, and the results were shown in Figure 4C.

We are very sorry for the loss of hydrogen bond between Ergotamine and TMPRSS2 due to the wrong abscissa in the process of making hydrogen bond interaction diagram.  In fact, both Lumacaftor and ergotamine had hydrogen bonds and strong interactions throughout the 100ns simulation.  The results are shown in Figure 4.  It is described in lines 136-138 of the article.

“As shown in Figure 4A and Figure 4B, during 100ns simulation, the number of hydrogen bonds formed by TMPRSS2 with Lumacaftor and ergotamine remained at 4 and 3, respectively.”

Point 5: Regarding figure 5, the authors should include in their analyses other interactions, not just H bonds. I suggest using LigPlot over the same representative structures.  That would highlight other relevant interactions that would then help explain the differences found in the MM-PBSA calculations. About the clustering (line 152), how many clusters were found? Are there any relevant differences, regarding ligand binding, amongst these structures?

Response 5: We are extremely grateful to reviewer for pointing out this problem. According to the reviewer's questions, representative structures were analyzed by Ligplot to obtain hydrogen bond and hydrophobic interaction data. See Figure 5. It is summarized in Table 1. About the clustering, 8 clusters are obtained for both complexes, which are described in lines 154-158 of the paper. The binding of ligands to proteins of these structures is shown in Figure S4 and Figure S5. The representative structures of these 8 clusters are similar and relatively stable.

“Subsequently, we categorized protein-ligand complex structures with similar conformations into the identical clusters during the 100 ns molecular simulation analysis. Eight clusters were found in both protein ligands.  Then ligand-protein binding of the representative structure of each cluster was observed, as shown in Figure S4 and Figure S5.”

Point 6: In Table 2, the rather large standard deviations suggest heterogeneity in the structures used for the calculation, in agreement with Figure 3A. That is not a sound protocol for MM-PBSA calculations.

Response 6: We are extremely grateful to reviewer for pointing out this problem. As for the question you asked, We recalculated the binding energy of the simulated data and found that the binding energy calculated in the first time was not accurate. We recalculated the new binding energy data and added them to Table 2.

Point 7: In the methods section, regarding the preparation of the structure, indeed, there are large gaps missing in the crystal structure. Reconstruction with SwissModel is a very good idea. On the other hand, there is a functional cleavage at residues 255 and 256, as stated in the UNIPROT entry for this protein, and I wonder if the authors took this into account. This processing can be important for the local electrostatics, as in generates extra N- and C-termini in the protein that could lie close to the active site, influencing ligand binding indirectly.

Response 7: Thank you very much for pointing out this problem. Thank you very much for pointing out this problem. We considered this, since the breaks of amino acids at position 255 and 256 are far away from the active pocket of the protein in 3D structure, and I also carefully read the literature and found that no one described them, so I think it will not have a great impact on the protein ligand binding.

References to similar articles are as follows:

  • In Silico Identification of Potential Natural ProductInhibitors of Human Proteases Key toSARS-CoV-2 Infection

  • Identification of novel TMPRSS2 inhibitors against SARS‑CoV‑2 infection: a structure‑based virtual screening and molecular dynamics study

  • Can polyoxometalates (POMs) prevent of coronavirus 2019-nCoV cell entry? Interaction of POMs with TMPRSS2 and spike receptor domain complexed with ACE2 (ACE2-RBD): Virtual screening approaches

  • Iterated Virtual Screening-Assisted Antiviral and Enzyme Inhibition Assays Reveal the Discovery of Novel Promising Anti-SARS-CoV-2 with Dual Activity

  • Computational screening of camostat and related compounds against human TMPRSS2: A potential treatment of COVID-19

Point 8: Regarding the MD simulations, a single 100 ns MD run to explore stability is not enough. See as a reference the work of Wonpil Im, where at least three runs are needed. Also, I would like to know why the SPC216 water model was used with CHARMM36, instead of the recommended TIP3P. Simulations with no added salt may lead to artificial increases in electrostatic interactions; as this is an extracellular protein, I suggest adding 0.15M NaCl.

Response 8: We are extremely grateful to reviewer for pointing out this problem. We have added two more simulations and put the RMSD for each simulation data into Figure S3.

About the water model issue, first of all, we do use the TIP3P water model when we get the topology file with the pdb2gmx command. Then, we use the editconf command to define the unit cell and fill it with the default spc216.gro. I think we caused confusion in writing this part and I apologize for that. Hence, we have modified this sentence in lines 240-241 as follow:

“The protein-ligand complexes are located in the center of the cubic cell and are solubilized using the TIP3P water model.”

Since our simulation is only a preliminary exploration, it is also found that the simulation results without salt are of certain reference significance by referring to relevant literature.  We will further explore the system with salt in the future.

Relevant references are listed below:

  • In Silico Identification of Potential Natural Product Inhibitors of Human Proteases Key to SARS-CoV-2 Infection

  • In-silico screening for identification of potential inhibitors against SARS-CoV-2 transmembrane serine protease 2 (TMPRSS2)

  • Can polyoxometalates (POMs) prevent of coronavirus 2019-nCoV cell entry? Interaction of POMs with TMPRSS2 and spike receptor domain complexed with ACE2 (ACE2-RBD): Virtual screening approaches

  • Computational screening of camostat and related compounds against human TMPRSS2: A potential treatment of COVID-19

Point 9: Finally, regarding MM-PBSA, I would like the authors to explain how the TS contribution is calculated, and also to state which section of the 100 ns run was used for the calculation. As I stated above, using the whole run is not adequate, due to the data for the ligands shown in figure 3A.

Response 9: We are extremely grateful to reviewer for pointing out this problem. TΔS is the entropy contribution, which can be obtained by normal mode analysis. However, this contribution is usually ignored in actual calculation. Because the system calculated by MM/PBSA scheme usually has little conformation change before and after the receptor-ligand binding, this contribution can be cancelled out in the calculation of the difference.  In addition, normal mode analysis is very time-consuming and has a large error limit, which introduces significant uncertainty. So we didn't calculate TΔS. The data for our MM/PBSA calculations are from the first 30ns of the MD simulation and are described in lines 176-178 of the paper.

“Due to the stability of the system, we chose the first 30ns for MM/PBSA calculation, and found in the simulation process that Lumacroft and Ergotamine are very tightly bound to TMPRSS2.”

Reviewer 3 Report

The submitted manuscript is about an "in silico" screening of already FDA-approved drugs to be repositioned for COVID-19 treatment. The authors proposed the TMPRSS2 as the target and screened a ZINC database. The results that they found to be their best were then evaluated through 100ns of molecular dynamics. No in vitro results are presented. The best result obtained from virtual screening (ansamycin r-116 - in Sup. Table 1, named as ansamethine r-116 in Line 108 of the manuscript) was discarded because "no information was found on the drug". Ansamycin is a class of antibiotics that comprises rifamycin and others. This specific molecule is not yet made available by the pharmaceutical industry, which could be a much better explanation to discard the substance. On the selected substances, ergotamine has already been related to SARS-CoV-2 inhibition by several other studies, not cited by the authors:

  • 10.1016/j.sjbs.2020.06.005
  • 10.1002/prot.26164 
  • 10.1080/07391102.2020.1794974
  • 10.1002/ptr.7442 (a review)
  • 10.2174/1566524021666210218113409
  • 10.1016/j.ejphar.2021.174082
  • 10.3906/biy-2012-52

All this available literature is mainly provided by in silico and in vitro results. A search by "ergotamine" and "covid" or "Sars-CoV-2" leads to many results, since ergotamine apparently can easily bind to several targets (from SARS-CoV-2 and human).

There was no mention, in the manuscript, of the toxicological profile of the substance. However, ergotamine in unlikely to be used together with protease inhibitors as ritonavir for example (since this association leads to ergotism). At the same time, nothing was discussed about the pharmacological profile of the substance. Vasoconstriction, the main effect of ergotamine (alpha-1 adrenergic agonist), can be lethal in severe covid conditions.

The other proposed ligand, lumacaftor, is a substance used to treat cystic fibrosis, in combination with ivacaftor. Besides its high price, other issues that were not discussed by the authors are present. This substance has, as the most frequent adverse effects, the development of dyspnea, abnormal respiration, and upper respiratory tract infection. 

Concomitant to these troubles, no in vitro results are presented. In the early pandemic era, only in silico results were interesting because of the lack of information and few laboratory-specific resources. Nowadays, both situations have been overcome. These "in silico" results could be interesting to publish in a Journal like "Molecules" if submitted with a virus permeability assay in cell culture at least. To justify the proposed action mechanism, binding or competitive assays are needed.

About the employed methodology, everything seems ok, but can be better explained (the grid box size choice or the solvation technique used in GROMACS for the MD, for example).

A good text review is needed as well to avoid things like "...Ergotamine/Lumacaftor systems after reaching equilibrium is reached..."(line 118) or "We also conclud the identical conclusion from..."

I suggest to the authors look forward to these experimental data and then resubmit the manuscript if the results are interesting. 

Author Response

Point 1: The submitted manuscript is about an "in silico" screening of already FDA-approved drugs to be repositioned for COVID-19 treatment.  The authors proposed the TMPRSS2 as the target and screened a ZINC database.  The results that they found to be their best were then evaluated through 100ns of molecular dynamics.  No in vitro results are presented.  The best result obtained from virtual screening (ansamycin r-116 - in Sup. Table 1, named as ansamethine r-116 in Line 108 of the manuscript) was discarded because "no information was found on the drug".  Ansamycin is a class of antibiotics that comprises rifamycin and others.  This specific molecule is not yet made available by the pharmaceutical industry, which could be a much better explanation to discard the substance.

Response 1: We are grateful for the suggestion. As suggested by the reviewer, we have changed the reasons for deprecating ansamycin R-116 in lines 109-112 of this article.

 “Ansamycin is a class of antibiotics, and this particular molecule is not currently used in the pharmaceutical industry. Therefore, the drugs Ergotamine and Lumacaftor, which had the second binding energy equally, were selected for the further molecular dynamics analysis.”

Point 2: On the selected substances, ergotamine has already been related to SARS-CoV-2 inhibition by several other studies, not cited by the authors:

  • 10.1016/j.sjbs.2020.06.005
  • 10.1002/prot.26164
  • 10.1080/07391102.2020.1794974
  • 10.1002/ptr.7442 (a review)
  • 10.2174/1566524021666210218113409
  • 10.1016/j.ejphar.2021.174082
  • 10.3906/biy-2012-52

All this available literature is mainly provided by in silico and in vitro results. A search by "ergotamine" and "covid" or "Sars-CoV-2" leads to many results, since ergotamine apparently can easily bind to several targets (from SARS-CoV-2 and human).

Response 2: We are extremely grateful to reviewer for pointing out this problem. As suggested by the reviewer, we have added this information to the discussion in lines 197 -200 and references 26-34, where the details are shown below:

“It has been proved that Ergotamine has good binding ability with SARS-CoV-2 MPRO, PLPRO, S protein, RdRp protein, 2 '-O-MTase and human NRP1[26-32]. Lumacaftor also has good binding ability with S protein[33, 34].”

Point 3: There was no mention, in the manuscript, of the toxicological profile of the substance. However, ergotamine in unlikely to be used together with protease inhibitors as ritonavir for example (since this association leads to ergotism). At the same time, nothing was discussed about the pharmacological profile of the substance. Vasoconstriction, the main effect of ergotamine (alpha-1 adrenergic agonist), can be lethal in severe covid conditions.

The other proposed ligand, lumacaftor, is a substance used to treat cystic fibrosis, in combination with ivacaftor. Besides its high price, other issues that were not discussed by the authors are present. This substance has, as the most frequent adverse effects, the development of dyspnea, abnormal respiration, and upper respiratory tract infection.

Response 3: We thank the reviewer for this critical question. As suggested by the reviewer, we have added this information to the discussion in lines 201-206.

“It is important to note that Ergotamine has some side effects as well as Lumacaftor. Common symptoms of Ergotamine include irritation, nausea, vomiting, headache, diarrhea, tingling in the limbs and confusion. The main effect of ergotamine (alpha-1 adrenergic agonist) is vasoconstriction, which can be very dangerous in severe cases of COVID. Lumacaftor also has some side effects. Dyspnea, abnormal breathing, and upper respiratory infections are common.”

Point 4: Concomitant to these troubles, no in vitro results are presented. In the early pandemic era, only in silico results were interesting because of the lack of information and few laboratory-specific resources. Nowadays, both situations have been overcome. These "in silico" results could be interesting to publish in a Journal like "Molecules" if submitted with a virus permeability assay in cell culture at least. To justify the proposed action mechanism, binding or competitive assays are needed.

Response 4: We are grateful for the suggestion. We also agree with you very much. We also believe that increasing in vitro experimental verification can better show the accuracy and completeness of experimental results. However, changchun, the city where I live, is suffering from COVID-19, and both the city and the laboratory are under lockdown, so the experimental work cannot be completed within a short period of time. We will complete this part of the work when the follow-up epidemic is over. Therefore, we also pointed out in the discussion that the following in vitro experiments will be the focus of our future research.

Point 5: About the employed methodology, everything seems ok, but can be better explained (the grid box size choice or the solvation technique used in GROMACS for the MD, for example).

A good text review is needed as well to avoid things like "...Ergotamine/Lumacaftor systems after reaching equilibrium is reached..."(line 118) or "We also conclud the identical conclusion from..."

I suggest to the authors look forward to these experimental data and then resubmit the manuscript if the results are interesting.

Response 5: We are grateful for the suggestion. The selection of the grid box size for the virtual screen is determined by re-docking the ligands in the crystal structure to the protein. Ensure that this box size is chosen so that the molecular docking results are closest to the crystal data.

We rewrote the sentence in lines 119-120 and lines 127-128.

“We performed 100 ns molecular dynamics simulations of the TMPRSS2 and ergotamine/Lumacaftor systems to understand their binding modes.”

“We also obtained the identical conclusion from the analysis of the radius of gyration (Rg) of the protein (Figure 3B).”

Reviewer 4 Report

In this paper, Wang et al report an in silico study on drug repurposing for TMPRSS2 inhibitors as possible candidates for SARS-CoV-2 infection treatment. The design of the study is classic using molecular docking, selection of top candidates and further investigation by molecular dynamics simulation and free energy calculations. Eventually, the authors report Lumacaftor and Ergotamine as the best candidates.

What I believe this study lacks is a comparison to known TMPRSS2 inhibitors. So far, the drug camostat is recognized as an inhibitor and there are even clinical trials at the moment. I would suggest additional calculations using structural similarity of top candidates towards camostat (both ligand and structure based) and estimated differences in binding energies.

Author Response

Point 1: What I believe this study lacks is a comparison to known TMPRSS2 inhibitors. So far, the drug camostat is recognized as an inhibitor and there are even clinical trials at the moment. I would suggest additional calculations using structural similarity of top candidates towards camostat (both ligand and structure based) and estimated differences in binding energies.

Response 1: We are grateful for the suggestion. As suggested by the reviewer, we added the docking data of camostat and TMPRSS2 molecule in supplementary Table 1. From the binding energy results, it is observed that camostat binds worse to the protein than the screened hit compounds. Therefore, we believe it may be more reasonable to focus on these hits compounds.

Round 2

Reviewer 2 Report

The authors addressed all the points I raised in my review, but, probably due to some misunderstanding, the following issues remain.

Regarding the supplementary figures, in Figure S1, which color corresponds to the crystal structure and which to the docked ligand? What is the RMSD for the ligands in the context of the complex?

In Figures S4 and S5, it is very hard to distinguish binding modes in this representation. I suggest to change these structures with the corresponding LigPlots. Then one can see the constellation of residues interacting with the ligands in each of the cluster centers.

In the Abstract, there remains the issue of having identified the active site, which the authors did not do (it is part of the PDB entry). I propose something like the following wording instead: "Based on the crystal structure, we targeted the active site of TMPRSS2 for virtual screening of compounds in the FDA database". This issue arises again in line 86.

In line 52, it is "active site", not "activation site".

The authors should uniformly decide whether compounds should be capitalized or not. 

In line 96, the rmsd between the docked and the xray structure should be stated. This is calculated for the ligand only, without superimposition.

In line 132, these are not "reaction regions", but binding regions.

Table 1 is hard to read and redundant with Figure 5. I suggest eliminating the table. It is worth noting that the ligands do not engage in interactions with the catalytic triad, but remain trapped at the entrance of the active site.

In line 178, a binding energy of ~-5 kcal/mol is not tight. Ergotamine is not a tight binder according to the MM-PBSA calculations; Lumacaftor is.

Lines 207-209 kill the value of the whole paper. These two compounds may be promising, but have serious side effects. Therefore, I suggest to propose them as lead compounds in order to improve both selectivity and pharmacokinetic properties.

Regarding the MM-PBSA calculations, the authors should include in the methods section the criteria used to select conformations for the free energy calculation.

Author Response

Response to Reviewer 2 Comments

Point 1: Regarding the supplementary figures, in Figure S1, which color corresponds to the crystal structure and which to the docked ligand? What is the RMSD for the ligands in the context of the complex?

Response 1: We are very grateful to the reviewers for their questions. As suggested by the reviewer, we have clarified the above problems in the annotation of Figure S1.

The annotations to Figure S1 are shown below:

Superposition of Nafamostat between conformation with lowest binding free energy and crystal structure. The green stick is Nafamostat in the crystal structure, and the blue stick is the structure of Nafamostat after docking. RMSD between them is 0.508 Å.

Point 2: In Figures S4 and S5, it is very hard to distinguish binding modes in this representation. I suggest to change these structures with the corresponding LigPlots. Then one can see the constellation of residues interacting with the ligands in each of the cluster centers.

Response 2: We are grateful for the suggestion. As suggested by the reviewer, the protein ligand interactions of each cluster representing the structure have been analyzed by LigPlots and shown in Figure S4 and Figure S5.

Point 3: In the Abstract, there remains the issue of having identified the active site, which the authors did not do (it is part of the PDB entry). I propose something like the following wording instead: "Based on the crystal structure, we targeted the active site of TMPRSS2 for virtual screening of compounds in the FDA database". This issue arises again in line 86.

Response 3: We are grateful for the suggestion. As suggested by the reviewer, we have modified lines 14-15 and 83-84.

Lines 14-15 read as follows:

Based on the crystal structure, we targeted the active site of TMPRSS2 for virtual screening of compounds in the FDA database.

Lines 83-84 read as follows:

The reconstructed structure and active center are shown in Figure 1.

Point 4: In line 52, it is "active site", not "activation site".

Response 4: We are extremely grateful to reviewer for pointing out this problem. According to the reviewer's questions, we have changed "Activation Site" to "Active Site"

Lines 48-50 read as follows:

Some important drug targets have been found: viral proteins and some host proteins, such as the viral spike protein, host cell ACE2 receptor, and TMPRSS2 active site

Point 5: The authors should uniformly decide whether compounds should be capitalized or not.

Response 5: We are extremely grateful to reviewer for pointing out this problem. According to the reviewer's questions, we have capitalized all compounds

Point 6: In line 96, the rmsd between the docked and the xray structure should be stated. This is calculated for the ligand only, without superimposition.

Response 6: Thank you very much for pointing out this problem. According to the reviewer's questions, we have described this in lines 92-94.

Lines 92-94 read as follows:

The superposition of Nafamostat between the lowest bound free-energy conformation and the crystal structure is shown in Figure S1, and RMSD between them is 0.508 Å (Figure S1).

Point 7: In line 132, these are not "reaction regions", but binding regions.

Response 7: Thank you very much for pointing out this problem. According to the reviewer's questions, we already made the substitution at line 131.

Lines 130-132 read as follows:

During the simulation, the binding regions of Ergotamine/Lumacaftor and protein were shown in Figure 3E and Figure 3F.

Point 8: Table 1 is hard to read and redundant with Figure 5. I suggest eliminating the table. It is worth noting that the ligands do not engage in interactions with the catalytic triad, but remain trapped at the entrance of the active site.

Response 8: We are extremely grateful to reviewer for pointing out this problem. According to the reviewer's questions, we have deleted Table 1 and modified the statements related to the interaction with catalytic triad.

Point 9: In line 178, a binding energy of ~-5 kcal/mol is not tight. Ergotamine is not a tight binder according to the MM-PBSA calculations; Lumacaftor is.

Response 9: We are extremely grateful to reviewer for pointing out this problem. We have recalculated the calculation of MM-PBSA of inhibitor and TMPRSS2 when we modified it last time, and generated the table. However, we forgot to update the table, so there was such a problem, for which I am deeply sorry. Now we have resubmitted the data in Table 1.

Point 10: Lines 207-209 kill the value of the whole paper. These two compounds may be promising, but have serious side effects. Therefore, I suggest to propose them as lead compounds in order to improve both selectivity and pharmacokinetic properties.

Response 10: We are extremely grateful to reviewer for pointing out this problem. According to the reviewer's questions, we have deleted lines 204-206 and modified them according to your comments

Lines 204-206 read as follows:

These two compounds may be promising for COVID-19, but have serious side effects. Therefore, we propose them as lead compounds to improve both their selectivity and pharmacokinetic properties.

Point 11: Regarding the MM-PBSA calculations, the authors should include in the methods section the criteria used to select conformations for the free energy calculation.

Response 11: We are extremely grateful to reviewer for pointing out this problem. According to the reviewer's questions, we have added this in the methods section.

Lines 266-268 read as follows:

We used g_mmpbsa to calculate the 100 ns trajectory obtained by MD simulation, and we selected the 0—30 ns trajectory after the system was stabilized to calculate the binding energy of the ligand and protein.

Reviewer 3 Report

.

Author Response

We are grateful for the suggestion. And we have extensively revised the manuscript in English through editorial services.

Reviewer 4 Report

My comments were properly addressed. The manuscript is now in a publishable form.

Author Response

We are grateful for the suggestion.